# Differential Modulation of Human M1 and M2 Macrophage Activity by ICOS-Mediated ICOSL Triggering

**DOI:** 10.3390/ijms24032953

**Published:** 2023-02-02

**Authors:** Casimiro Luca Gigliotti, Chiara Dianzani, Ian Stoppa, Chiara Monge, Salvatore Sutti, Daniele Sblattero, Chiara Puricelli, Roberta Rolla, Umberto Dianzani, Elena Boggio

**Affiliations:** 1Department of Health Sciences, Università del Piemonte Orientale, 28100 Novara, Italy; 2NOVAICOS s.r.l.s, Via Amico Canobio 4/6, 28100 Novara, Italy; 3Department of Scienza e Tecnologia del Farmaco, University of Turin, 10125 Turin, Italy; 4Department of Life Sciences, University of Trieste, 34127 Trieste, Italy; 5Clinical Biochemistry Laboratory, Maggiore della Carità University Hospital, 28100 Novara, Italy

**Keywords:** macrophage polarization, ICOS:ICOSL system, anti-inflammatory activities

## Abstract

Activated T cells express the inducible T-cell co-stimulator (ICOS) that, upon binding to its ubiquitously expressed ligand (ICOSL), regulates the immune response and tissue repair. We sought to determine the effect of ICOS:ICOSL interaction on human M1 and M2 macrophages. M1 and M2 macrophages were polarized from monocyte-derived macrophages, and the effect of a soluble recombinant form of ICOS (ICOS-CH3) was assessed on cytokine production and cell migration. We show that ICOS-CH3 treatment increased the secretion of CCL3 and CCL4 in resting M1 and M2 cells. In LPS-treated M1 cells, ICOS-CH3 inhibited the secretion of TNF-α, IL-6, IL-10 and CCL4, while it increased that of IL-23. In contrast, M2 cells treated with LPS + IL4 displayed enhanced secretion of IL-6, IL-10, CCL3 and CCL4. In CCL7- or osteopontin-treated M1 cells, ICOS-CH3 boosted the migration rate of M1 cells while it decreased that of M2 cells. Finally, β-Pix expression was upregulated in M1 cells and downregulated in M2 cells by treatment with ICOS-CH3. These findings suggest that ICOSL activation modulates the activity of human M1 and M2 cells, thereby eliciting an overall anti-inflammatory effect consistent with its role in promoting tissue repair.

## 1. Introduction

The inducible T-cell co-stimulator (ICOS, CD278) is a receptor mainly found on activated T cells and, at lower levels, on human dendritic cells (DCs) and plasma cells [1]. ICOS binds to its specific ligand ICOSL (CD275, also known as B7h, B7-H2, GL-50, or B7RP1) expressed by several cell types, such as DCs, macrophages, B cells, endothelial cells (ECs), epithelial cells, and fibroblasts [2]. ICOSL-mediated triggering of ICOS regulates T-cell activation in lymphoid organs and T-cell functions at inflamed sites by promoting the differentiation of regulatory T (Treg) and T helper (Th)-17 cells. Moreover, being highly expressed in T follicular helper (TFH) cells, ICOS regulates the development of germinal centers, and its deficiency causes common variable immunodeficiency. Lastly, ICOS plays a role in response to tumors and intracytoplasmic pathogens in CD8^+^ T cells [3,4,5,6,7].

The interaction between ICOS and ICOSL can also result in reverse signaling, whereby ICOSL activation by ICOS modulates the activity of several ICOSL-expressing cell types. For instance, ICOS co-stimulation increased Interleukin (IL)-6 secretion in mouse DCs [8]. Furthermore, in human cells, we have shown that activation of ICOSL with a recombinant soluble construct of ICOS (ICOS-Fc) can lead to the following effects: (i) impairment of EC adhesiveness and migration in vitro [9,10]; (ii) modulation of cytokine secretion, promotion of antigen cross-presentation, and inhibition of adhesiveness and migration of DCs in vitro [11]; (iii) decrease in adhesiveness and migration rate of tumor cell lines in vitro and development of experimental lung metastases in vivo [10]; (iv) inhibition of osteoclast differentiation and bone resorption in vitro and development of osteoporosis in vivo [12]; (v) downregulation of phospho-Erk1,2 and -p38 in ECs, dephosphorylation of Focal adhesion kinase (FAK) in tumor cells, and downregulation of β-Pix in DCs and tumor cells [9,10,11]; and (vi) growth inhibition of established primary tumors through angiogenesis impairment—in these experiments ICOS-Fc was encapsulated in biocompatible and biodegradable nanoparticles [13,14].

Another important function of ICOSL relies on its ability to bind to osteopontin (OPN), a protein that can act as both a component of the extracellular matrix and a pro-inflammatory cytokine [15]. Of note, ICOS and OPN bind to ICOSL at different non-overlapping sites, eliciting distinct—often opposite—activities. In particular, tumor angiogenesis and metastasization and migration of ECs, DCs and tumor cells are promoted by OPN and inhibited by ICOS.

More recently, we have shown that wounded mice treated with ICOS-Fc display enhanced tissue repair and that this effect partly involves tissue-repairing macrophages [16]. According to a well-established model, classical activation of macrophages (LPS + IFNγ) generates M1 cells endowed with pro-inflammatory properties, while their alternative activation with IL-4 gives rise to M2 macrophages involved in tissue repair.

On the one hand, M1 macrophages are not only responsible for pathogen clearance through the generation of reactive oxygen species (ROS), but they also mediate tissue damage and curb tissue regeneration and wound healing. In addition, M1 cells exert a potent pro-inflammatory activity by secreting cytokines such as IL-1 and Tumor Necrosis Factor (TNF)-α, and their development is supported by Th1-type cytokines, such as Interferon (IFN)γ and TNF-α. On the other hand, M2 macrophages exert anti-inflammatory activity and favor tissue repair by secreting cytokines such as IL-10. Depending on the activating stimulus, M2 macrophages can be divided into four subtypes: (1) M2a (IL-4 and IL-13); (2) M2b (immune complexes and Toll-like receptor (TLR) agonists or IL-1 receptor ligands); (3) M2c (glucocorticoids and IL-10); and (4) M2d (TLR agonists and adenosine receptors) [17,18,19].

Our recent studies in mouse models of acute liver damage and skin wound healing [16,20] have demonstrated that ICOS-mediated triggering of ICOSL promotes tissue repair through the recruitment of M2-like macrophages to the wound site. Moreover, ICOS-mediated activation of ICOSL inhibits the migration rate of mouse M1 cells induced by C-C motif chemokine ligand (CCL) 2 or OPN in vitro, whereas it increases that of similarly treated M2 cells [16]. This finding was unexpected as we had previously shown that ICOSL activation by ICOS inhibited cell migration in ECs, DCs, and several types of tumor cell lines [9,10,11]. Thus, the aim of this work was to assess the effects of ICOS-mediated triggering of ICOSL on human M1 vs. M2 macrophages. To this end, we took advantage of a fusion protein of the extracellular portion of the human ICOS fused to the human CH3 domain of the IgG_1_ Fc portion (ICOS-CH3), capable of activating ICOSL but not FcγRs.

Our results show that ICOS-CH3 modulates cytokine secretion in human M1 and M2 cells, with an overall effect of decreasing the effector phase of inflammation and promoting tissue repair. Intriguingly, analysis of cell migration reveals an opposite pattern from that seen in mouse M1 and M2 cells since ICOS-mediated triggering of ICOSL inhibits CCL7- or OPN-induced migration of human M2 cells, whereas it increases that of similarly treated M1 cells.

## 2. Results

### 2.1. Analysis of ICOSL and ICOS Expression in M1 vs. M2 Cells

Monocytes were induced to differentiate monocyte-derived macrophages (MDMs) by culturing them with GM-CSF (resting M1 cells) or M-CSF (resting M2 cells) for 6 days (T6). M1 cells were then activated by growing them, for 2 additional days (T6 + 2), in the presence of lipopolysaccharide (LPS) or IFNγ or LPS + IFNγ since LPS and IFNγ are typical pro-inflammatory stimuli of M1 cells. M2 cells were instead cultured with LPS or IL-4 or LPS + IL-4 since LPS and IL-4 are involved in the polarization of different subtypes of M2 cells [17,18,19]. Finally, the surface expression of ICOSL and ICOS was evaluated in resting (T6) vs. activated (T6 + 2) cells. We found that both M1 and M2 cells robustly expressed ICOSL (Figure 1a–c), while ICOS expression levels were barely detectable (Figure 1b–d). M2 cells expressed higher levels of ICOSL than M1 cells in any culture conditions (Figure 1a–c), whereas ICOS expression in M1 cells was similar to that of M2 cells (Figure 1b–d). Activation of M1 cells always led to the downregulation of ICOSL expression, regardless of the stimulus (Figure 1a). In M2 cells, ICOSL expression was downregulated following activation with LPS or, to a lesser extent, LPS + IL-4, but not IL-4 alone (Figure 1c). ICOS expression remained unchanged in activated M2 (Figure 1d), whereas it displayed a slight upregulation in M1 cells activated with LPS but not in those activated with LPS + IFNγ or IFNγ alone (Figure 1b). Two color staining showed that the few cells expressing ICOS co-expressed ICOSL in any culture condition (Figure 1e,f).

### 2.2. Effects of ICOSL Triggering on Cytokine Secretion

To assess the functional effect of ICOSL triggering, we used the recombinant ICOS-CH3 molecule, which triggers ICOSL but not FcγRs. The point mutation variant ^F119S^ICOS-CH3, which does not bind to ICOS, was used as a negative control. Initial experiments in resting cells showed that, as expected, M1 cells expressed higher levels of CD80 and CD86, marking M1 cells, and lower levels of CD163, marking M2 cells, than M2 cells (Appendix A).

We next analyzed the effects of ICOS-CH3 on the secretion of IL-6, IL-23 and TNF-α, mainly produced by M1 cells, and IL-10, mainly produced by M2 cells. Moreover, we assessed the expression of CCL3 and CCL4, which play a key role in the recruitment of inflammatory cells into damaged tissues. For this purpose, M1 cells were activated with LPS, IFNγ, or LPS + IFNγ, whereas M2 cells were exposed to LPS, IL-4, or LPS + IL-4 in the presence or absence of ICOS-CH3, and their supernatants were collected after 48 h (T6 + 2) to determine cytokine secretion by ELISA. The pattern of cytokine secretion in the absence of ICOS-CH3 is shown in Appendix A, and the expression of CD80, CD86, and CD163 is in Appendix A.

The expression of CD86 did not substantially change in the different activation conditions whereas, in M2 cells, the expression of CD80 was increased by treatments with LPS or LPS + IL-4, while CD163 was decreased by treatment with LPS or IL-4 or LPS + IL-4.

Resting M1 and M2 cells and those cultured in the presence of IFNγ (M1 cells) or IL-4 alone (M2 cells) produced low levels of all cytokines. Resting M2 cells produced higher levels of IL-10 compared to those secreted by resting M1 cells. M1 cells cultured in the presence of IFNγ produced higher levels of IL-6 and lower levels of IL-10 compared to IL-4-treated M2 cells. By contrast, activation with LPS in the presence or absence of IFNγ (M1 cells) or IL-4 (M2 cells) strikingly increased cytokine production, commensurate with the expected differences between the two cell types. In the presence of LPS alone, M1 cells produced higher levels of TNF-α, IL-6, and IL-23 than those seen in M2 cells, which in turn generated higher levels of IL-10. Both cell types secreted similar high levels of CCL3 and CCL4. In the presence of LPS plus either IFNγ (M1) or IL-4 (M2), we obtained similar results except for IL-10, whose difference between M1 and M2 cells was lost.

Analysis of the effect of ICOS-CH3 on the low cytokine levels produced in the absence of LPS revealed that only CCL3 and CCL4 secretion was upregulated in both M1 and M2 cells, either resting or stimulated with IFNγ (M1) or IL-4 (M2) (Figure 2). On the other hand, no effect was detected with regard to the other cytokines [21]. Treatment with ^F119S^ICOS-CH3 had no effects under any culture conditions.

In M1 cells treated with LPS alone, ICOS-CH3 reduced the secretion of IL-10, IL-6, TNF-α, and CCL4 while it increased that of IL-23. When IFNγ was added to these mixtures, we could no longer observe any effect of ICOS-CH3 on IL-10, IL-6, and IL-23 secretion, while the inhibitory effect on TNF-α and CCL4 secretion was retained. In addition, it appeared to have an inhibitory effect on CCL3 secretion. Treatment with ^F119S^ICOS-CH3 had no effect under any experimental conditions (Figure 3).

In LPS-treated M2 cells, ICOS-CH3 reduced TNF-α and CCL3 secretion. Conversely, in LPS + IL-4 treated M2 cells, ICOS-CH3 increased secretion of IL-10, IL-6, CCL3 and CCL4, whereas IL-23 was always undetectable [21]. Treatment with ^F119S^ICOS-CH3 did not exert any effect under any stimulatory conditions (Figure 4).

### 2.3. Effect of ICOSL Stimulation on Cell Migration

To investigate the effect of ICOSL triggering on M1 and M2 cell migration, we performed a Boyden chamber migration assay using CCL7 as a chemoattractant. In some experiments, we used OPN as a chemoattractant because we previously showed that its chemotactic activity depends on the cellular expression of ICOSL [15]. Cells previously activated with different stimuli (i.e., LPS, IFNγ, LPS + IFNγ for M1 cells; LPS, IL-4, LPS + IL-4 for M2 cells) were seeded in the upper chamber of the transwell apparatus in the presence or absence of ICOS-CH3 or ^F119S^ICOS-CH3. The lower chamber was loaded with medium with or without CCL7 or OPN, and cell migration in the lower chamber was assessed after 6 h. In the absence of ICOS-CH3 or ^F119S^ICOS-CH3, we found that both chemoattractants triggered a substantial migration of both M1 and M2 cells under any conditions, albeit with a few minor differences. IL-4-activated M2 cells showed a higher migration rate than that of IFNγ-activated M1 cells in response to CCL7. A similar trend (*p* = 0.06) was observed following OPN exposure. In the case of cells activated with LPS alone, M2 cells showed a higher migration rate than that of M1 cells in response to OPN, whereas the opposite pattern was detected following stimulation with CCL7. On the other hand, no significant difference in migration rates was observed between cells activated with LPS plus IFNγ (M1 cells) or IL-4 (M2 cells) (Appendix A).

The addition of ICOS-CH3, but not ^F119S^ICOS-CH3, to the upper chamber led to robust induction of CCL7- or OPN-driven M1 cell migration, while it inhibited that of M2 cells under any experimental conditions (Figure 5).

### 2.4. Effect of ICOSL Stimulation on Downstream Signaling Pathways

To investigate the effect of ICOSL triggering on downstream signaling pathways, we treated LPS-activated M1 and M2 cells (T6 + 2) with either medium, ICOS-CH3_,_ or ^F119S^ICOS-CH3 (10 μg/mL each) for 30 min, and then assessed the expression level of β-Pix, that was previously found to be downregulated by ICOSL triggering in different cell types [9,10,11,12]. Expression of β-Pix was upregulated in M1 cells and downmodulated in M2 cells by treatment with ICOS-CH3; ^F119S^ICOS-CH3 had no effects (Figure 6).

## 3. Discussion

In this study, we show that activation of ICOSL with the soluble recombinant ICOS-CH3 molecule differentially modulates cytokine secretion and cell migration of human M1 vs. M2 macrophages cultured in different activation conditions.

In resting M1 and M2 macrophages, treatment with ICOS-CH3 mainly induces secretion of CCL3 and CCL4, regardless of the presence of IFNγ or IL-4 enhancing M1 and M2 differentiation. On the other hand, the secretion of the effector cytokines driving inflammation does not seem to be substantially influenced. In contrast, ICOS-CH3 exposure of LPS-activated M1 and M2 cells decreases TNFα secretion in M1 cells, whereas it increases that of IL-10 in M2 cells, thus resulting in an overall anti-inflammatory effect. Noteworthy, this stimulation also augments IL-23 secretion in M1 cells, which is in line with previous data obtained in DCs [11]. These findings are also in good agreement with the observation that ICOS-CH3 stimulation of resting M1 and M2 increases CCL3 and CCL4 secretion, which would then be expected to enhance the recruitment of polymorphonuclear cells and other immune cells. Indeed, it is well established that IL-23 supports IL-17-mediated type-3 inflammatory response, thus promoting the secretion of IL-8-driven neutrophil recruitment [22,23]. Therefore, it is possible that T cells infiltrating inflamed tissues may orchestrate the local inflammatory response by not only producing cytokines but also triggering ICOSL signaling on macrophages. In the first phase, Th1 and Th17 inflammatory T cells would coordinate the defensive response by both producing inflammatory cytokines (i.e., IFNγ and IL-17) and triggering ICOSL on M1 cells, resulting in increased secretion of IL-23, ultimately favoring a type 3 response and neutrophil recruitment. In a second phase, Th2 and Treg cells would cool down the defensive response and coordinate tissue repair by both producing anti-inflammatory cytokines (i.e., IL-4, IL-10, and TGFβ) and activating ICOSL on M2 cells, resulting in increased secretion of IL-10 and CCL3, which again would promote the recruitment of neutrophils involved in the removal of tissue debris and the release of pro-angiogenic factors [24]. In both phases, neutrophil recruitment would also be supported by activating ICOSL in bystander resting macrophages, leading to increased CCL3 and CCL4 secretion.

Another intriguing observation is the opposite modulation of IL-6 secretion by ICOS-CH3 in M1 vs. M2 cells, in light of the fact that IL-6 can exert both pro- and anti-inflammatory effects. Specifically, IL-6 secretion was 8-fold higher in M1 cells activated with LPS than in similarly activated M2 cells; and it was 36-fold higher in M1 cells activated with LPS + IFNγ than in M2 cells activated with LPS + IL-4. Furthermore, ICOS-CH3 decreased IL-6 secretion by about 60% in M1 cells activated with LPS, while it increased it by about 2.5 folds in M2 cells activated with LPS + IL-4. On the one hand, inhibition of IL-6 secretion in M1 cells is in line with the global anti-inflammatory effect of ICOS-CH3 in these cells, raising the attractive possibility that, in an in vivo setting, ICOS-CH3 infusion might even decrease circulating IL-6, thus inhibiting the production of acute phase proteins in the liver [25]. Fittingly, we have recently reported that, in a mouse model of sepsis, ICOS-deficient mice display a striking increase in blood IL-6 levels, which are substantially decreased by ICOS-Fc treatment [26]. On the other hand, enhanced IL-6 secretion induced by ICOS-CH3 in M2 cells is in keeping with the involvement of these cells in tissue repair, where IL-6 supports the formation of new vessels [27]. In good agreement, we have recently shown that ICOS- and ICOSL-deficient mice display delayed wound healing with decreased IL-6, VEGF, and angiogenesis in the healing tissue and that all these defects are counteracted by topic treatment with ICOS-Fc, which increases infiltration of M2-like macrophages as well [16].

A global anti-inflammatory and pro-repair activity of ICOS-mediated ICOSL triggering in macrophages are also supported by our recent in vivo study showing that CCl_4_-induced liver damage, mediated by massive infiltration of blood-borne macrophages, is dramatically worsened in mice deficient for either ICOS or ICOSL due to decreased number of ICOS^+^ CD8^+^ T cells and reparative MDMs in the liver. Conversely, in ICOS-(Knock-out) KO but not wild-type or ICOSL-KO mice, treatment with ICOS-Fc protects from liver toxicity and increases the number of reparative MDMs in the liver [20].

Another interesting point of this study is the differential effect of ICOS-CH3 on the migratory responses of M1 and M2 cells to CCL7 and OPN since ICOS-CH3 treatment increased the migration rate of M1 cells, but it reduced the migration of M2 cells. The fact that M1 and M2 macrophages display different migration behaviors is not surprising *per se*, as they have been reported to display different migration properties in both mice and humans in several settings [28,29,30]. What is surprising is the observation that human M1 and M2 cells display an opposite response pattern to ICOS-CH3 from the one seen in their murine counterparts. Indeed, we have recently reported that, in mouse cells, CCL2- or OPN-driven migration of M2 cells is increased following stimulation with a recombinant murine ICOS-Fc-fusion protein, unable to bind mouse FcγRs, while it was decreased in similarly stimulated M1 cells [16]. The opposite effect of ICOSL triggering on the migration of M1 and M2 cells in humans vs. mice resembles the opposite activity of ICOS on the production of Th1 and Th2 cytokines in these two species since ICOS triggering mainly increases secretion of IL-4 in mice, whereas it increases that of IFNγ in humans [6,7,31].

This study and previous works [9,10,11] suggest that ICOS-CH3 effects on cell migration might involve β-Pix, which was upregulated in M1 cells and downregulated in M2 cells (Figure 6), which fits with the previous data showing that ICOSL triggering reduces cell migration and β-Pix expression in DCs and tumor cell lines.

## 4. Materials and Methods

### 4.1. Cells

Peripheral blood mononuclear cells (PBMCs) were separated from buffy coats provided by the local Blood Transfusion Service (Novara, Italy) by density gradient centrifugation using Ficoll-Hypaque (Limpholyte-H, Cedarlane Laboratories, Burlington, ON, Canada). The use of buffy coats was approved by the local Ethics Committee (No. CE 88/17), and the study was conducted in accordance with the Declaration of Helsinki. MDMs were prepared from CD14^+^ monocytes isolated with the EasySep^TM^ Human CD14 Negative Selection Kit (StemCells Technologies, Vancouver, BC, USA). Monocytes (0.5 × 10^6^) were plated in 6-well plates and cultured for 6 d in DM composed of RPMI-1640 (Thermo Fisher Scientific, Waltham, MA, USA), 2 mM L-glutamine, 10% FBS (Thermo Fisher Scientific), 25 mM Hepes, 100 U/mL penicillin, and 100 μg/mL streptomycin, and recombinant human GM-CSF for M1 polarization (100 ng/mL; R&D System, Minneapolis, MN, USA) or M-CSF for M2 polarization (100 ng/mL; R&D Systems). DM was changed every 3 d. As for the maturation assay, MDMs were cultured for 2 additional days in DM in the presence of LPS (1 µg/mL, *Escherichia coli*, serotype 055:B5, Sigma-Aldrich, St Louis, MO, USA)—with or without IFNγ (50 ng/mL, R&D System) for M1 or IL-4 (10 ng/mL, R&D System) for M2—and ICOS-CH3 (5 µg/mL) produced in house. ICOS-CH3 is a fusion protein of the extracellular portion of the human ICOS fused to the human CH3 domain of the IgG_1_ Fc portion [32]. The ^F119S^ICOS-CH3 mutant, carrying the F119S substitution in the ICOS amino acid sequence, unable to bind to ICOS, was used as a negative control.

### 4.2. Immunofluorescence

The M1 and M2 MDM surface phenotype was assessed by flow cytometry (BD Biosciences, San Diego, CA, USA) using Fluorescein isothiocyanate (FITC)-, Phycoerythrin (PE)-, Peridinin chlorophyll protein-Cyanine5.5 (PercP/Cy5.5)- and, Allophycocyanin (APC) conjugated mAb to CD80, CD86, CD163 (BioLegend, San Diego, CA, USA), ICOSL and ICOS (R&D Systems).

### 4.3. Cytokine and Chemokine Analysis

Secretion of IL-6, IL-10, IL-23, TNF-α, CCL-3, and CCL-4 was evaluated in the culture supernatant by standard enzyme-linked immunosorbent assays (ELISA) (R&D Systems and eBioscience, San Diego, CA, USA).

### 4.4. Migration Assay

In order to determine the effects of ICOS on MDM migration, we used the Boyden chamber migration assay (BD Biosciences). Differentially activated M1 or M2 cells were plated (10^4^ or 2 × 10^4^ cell/well) into the apical side of 50 µg/mL Matrigel-coated filters in serum-free medium in the presence and absence of ICOS-CH3 (5 µg/mL) or ^F119S^ICOS-CH3 (5 µg/mL). Medium containing human CCL7 (30 nM, Immunotools, GmbH, Friesoythe, Germany) or human osteopontin (OPN, 10 μg/mL, R&D System) was used as a chemoattractant. After 6 h, the cells on the apical side were wiped off with Q-tips. The cells on the bottom of the filter were stained with crystal violet and counted (quadrupled filter) with an inverted microscope. Data are shown as percentages of migrating treated cells relative to that of migrating untreated cells, as previously reported [33].

### 4.5. Immunoblotting

M1 and M2 MDM were lysed in 50 mM of Tris-HCl (pH 7.4), 150 mM of NaCl, 5 mM of EDTA, and 1% Nonidet P-40 with phosphatase and protease inhibitor cocktails (Sigma-Aldrich). Then, 20 μg of proteins were run on 10% SDS-PAGE gels and transferred onto Hybond-C extra nitrocellulose membranes (GE Healthcare, Piscataway, NJ, USA). The membranes were then probed with Abs to anti-β-Pix (AdipoGen Life Sciences, San Diego, CA, USA) and β-actin (Sigma-Aldrich), followed by horseradish peroxidase (HRP) conjugated secondary Ab (Sigma-Aldrich). The bands were detected by chemiluminescence using the VersaDoc Imaging System (Bio-Rad Laboratories, Hercules, CA, USA).

### 4.6. Statistical Analysis

Results were analyzed using the Wilcoxon or paired T-test and Mann-Whitney or unpaired T-test using the GraphPad Instat3 software (San Diego, CA, USA), and the significance was set at *p* < 0.05 (two-tailed test).

## 5. Conclusions

In conclusion, this work provides a new paradigm of T-cell regulation of macrophage functions based on ICOS:ICOSL-mediated modulation of human M1 and M2 cell migration and cytokine secretion that could ultimately be exploited for the design of novel drugs that can modulate tissue repair. In acute tissue damages, ICOS-CH3-like drugs might support tissue repair by decreasing TNF-α and increasing IL-23 secretion in M1 cells, while increasing IL-10 secretion in M2 cells and balancing IL-6 secretion in the two cell types. Moreover, these drugs might increase the migration and recruitment of M1 cells into the damaged tissue and favor their reprogramming to M2 cells driven by the anti-inflammatory environment. In turn, inhibition of M2 cell migration might favor their stay in the repairing tissue (Figure 7).

## Figures and Tables

**Figure 1 ijms-24-02953-f001:**
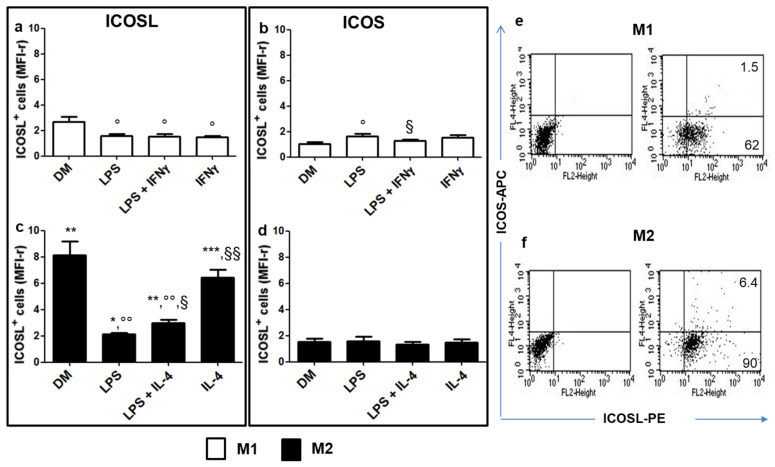
ICOSL and ICOS expression on M1 and M2 cells. Monocytes were cultured in a differentiation medium (DM) (GM-CSF for M1 polarization or M-CSF for M2 polarization). ICOSL (**a**–**c**) and ICOS (**b**–**d**) expression was evaluated after 6 days (**d**) (T6) of culture in the presence of GM-CSF or M-CSF alone or after an additional 48 h (T6 + 2) in the presence of LPS, IFNγ or LPS + IFNγ or LPS, IL-4, or LPS + IL-4 (M2) by flow cytometry. Data represent the mean and SEM expressed as mean fluorescence intensity ratio (MFI-R) of results from five independent experiments. Representative cytofluorimetric plots of ICOS and ICOSL expression in DM-treated cells are shown in panels (**e**) (M1 cells) and (**f**) (M2 cells); left panels: negative controls, right panels: positive staining. MFI-R was calculated considering all viable cells according to the following formula: MFI of the stained sample histogram (arbitrary units)/MFI of the control histogram (arbitrary units). * *p* < 0.05, ** *p* < 0.01, *** *p* < 0.0001 versus the corresponding M1; § *p* < 0.05, §§ *p* < 0.01 vs LPS or ° *p* < 0.05, °° *p* < 0.01 vs. DM.

**Figure 2 ijms-24-02953-f002:**
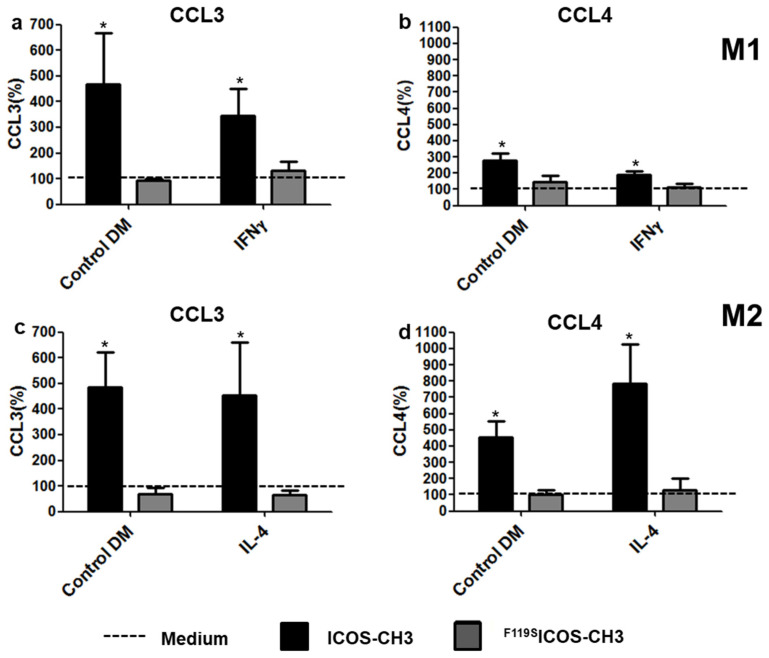
Chemokine secretion of M1 and M2 cells in the absence of LPS. M1 (**a**,**b**) or M2 (**c**,**d**) MDMs were cultured in differentiation media (DM) in the presence or absence of IFNγ IL-4 plus ICOS-CH3 or ^F119S^ICOS-CH3. Culture supernatants were harvested after 2 d and examined for CCL3 and CCL4 production by ELISA. Data represent the mean and SEM expressed as a percentage of results from six independent experiments. The corresponding samples cultured in the absence of ICOS-CH3 or ^F119S^ICOS-CH3 were set at 100% (whose absolute data are shown in Appendix A). * *p* < 0.05 vs. the corresponding 100% control.

**Figure 3 ijms-24-02953-f003:**
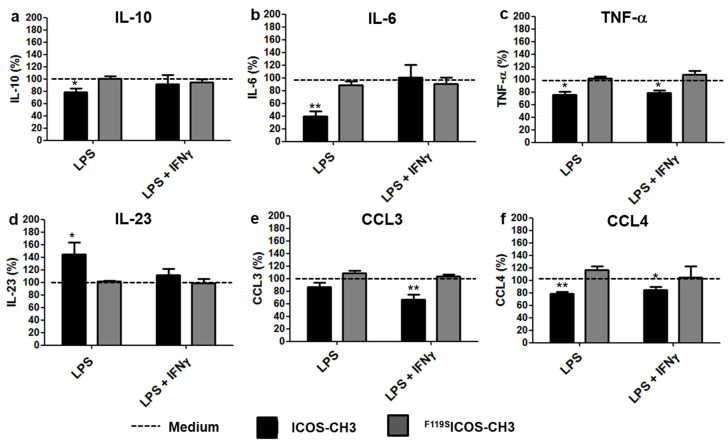
Effect of ICOS-CH3 on cytokine/chemokine secretion of M1 cells in the presence of LPS. M1 MDMs were cultured in DM + LPS in the presence or absence of IFNγ plus ICOS-CH3 or ^F119S^ICOS-CH3. Culture supernatants were harvested after 2 d and examined for IL-10 (**a**), IL-6 (**b**), TNF-α (**c**), IL-23 (**d**), CCL3 (**e**), and CCL4 (**f**) production by ELISA. Data represent the mean and SEM expressed as a percentage of results from five/six independent experiments. The corresponding samples cultured in the absence of ICOS-CH3 or ^F119S^ICOS-CH3 were set at 100% (whose absolute data are shown in Appendix A). * *p* < 0.05, ** *p* < 0.01 vs. the corresponding 100% control.

**Figure 4 ijms-24-02953-f004:**
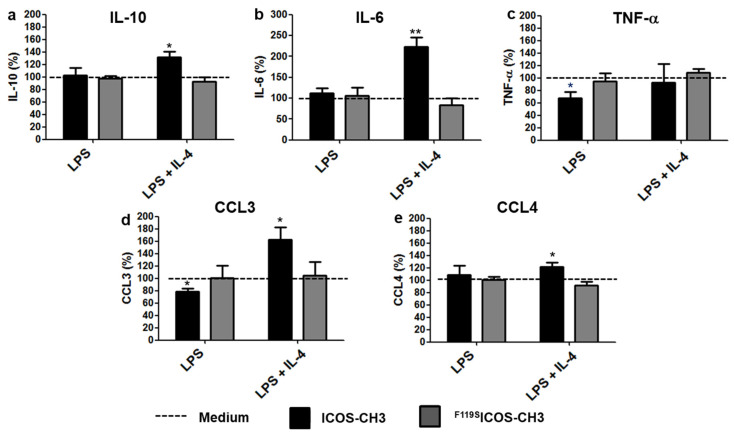
Effect of ICOS-CH3 on cytokine/chemokine secretion of M2 cells treated with different stimuli. M2 MDMs were cultured with DM + LPS in the presence or absence of IL-4 plus ICOS-CH3 or ^F119S^ICOS-CH3. Culture supernatants were harvested after 2 d and examined for IL-10 (**a**), IL-6 (**b**), TNF-α (**c**), CCL3 (**d**), and CCL4 (**e**) production by ELISA. Data represent the mean and SEM expressed as a percentage of results from five/six independent experiments performed in triplicate. The corresponding samples cultured without ICOS-CH3 or ^F119S^ICOS-CH3 were set at 100% (whose absolute data are shown in Appendix A). * *p* < 0.05, ** *p* < 0.01 vs. the corresponding 100% control.

**Figure 5 ijms-24-02953-f005:**
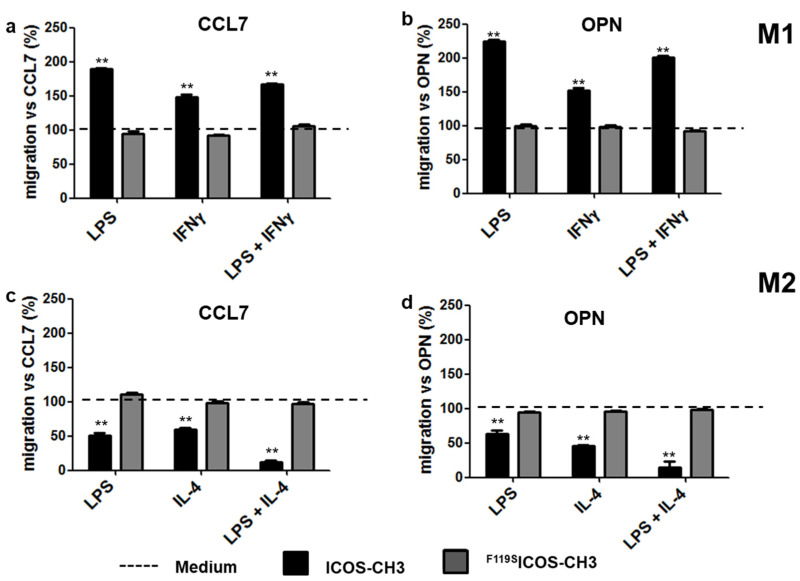
Effect of ICOS-CH3 on M1 and M2 cell migration. (**a**,**b**) M1 MDMs were treated with LPS, IFNγ, or LPS + IFNγ and (**c**,**d**) M2 MDMs with LPS, IL-4, or LPS + IL-4. Cells were plated on the apical side of Matrigel-coated filters in the presence or absence of ICOS-CH3 or ^F119S^ICOS-CH3. The basolateral chamber was loaded with a medium in the presence of either (**a**–**c**) CCL7 (30 nM) or (**b**–**d**) OPN (10 µg/mL). After 6 h, the cells migrated to the bottom of the filters, were stained using crystal violet and counted using an inverted microscope. Data are expressed as the percentage of migrating cells under each condition and are represented as the mean and SEM of the results from 3 to 8 experiments performed in quadruplicate. Basal migration in the presence of either CCL7 or OPN was set at 100%. ** *p* < 0.01 vs. CCL7 or OPN control.

**Figure 6 ijms-24-02953-f006:**
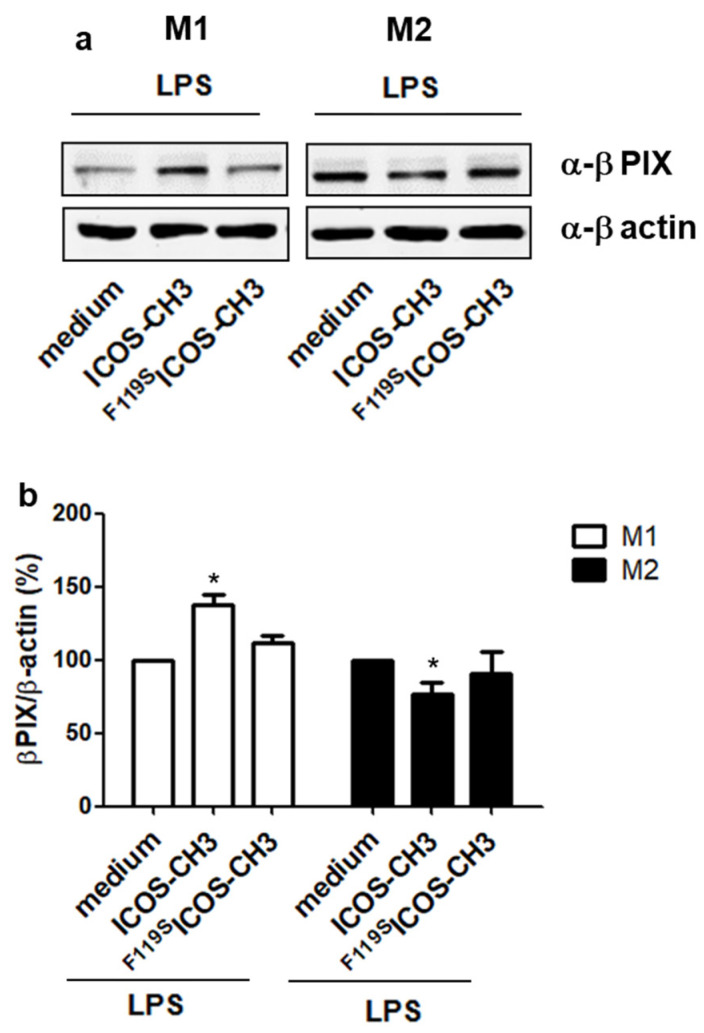
Downstream effects of ICOSL activation in M1 vs. M2 cells. Differentiated M1 and M2 MDMs were treated or not with the indicated ICOS reagents for 30 min at T6 + 2. Expression of β-Pix was assessed by Western blot. The same blots were also probed with an anti–β-actin Ab as a loading control. The upper panel (**a**) shows a representative experiment. The bar graph (**b**) shows the densitometric analyses of the gels referred to as the relative internal control. Data are expressed as mean, and SEM of the percentage of increase *vs*. control set at 100% from three independent experiments. * *p* < 0.05, vs. control.

**Figure 7 ijms-24-02953-f007:**
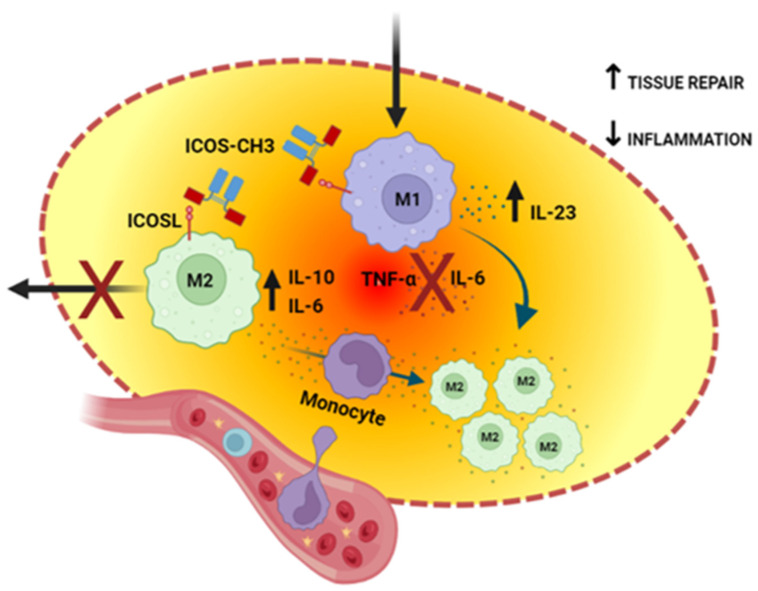
Graphical representation of the ICOS-CH3 differential effects of M1 and M2 cells in the setting of acute tissue damage. ICOS-CH3 inhibits the secretion of TNF-α and IL-6 in M1 cells while promoting IL-10 secretion in M2 macrophages. Moreover, it balances IL-6 secretion, increases in M2 cells and decreases in M1 cells. The ICOS-CH3 effect on cell migration would increase the recruitment of M1 cells, which may be reprogrammed to M2 cells by the surrounding anti-inflammatory environment while retaining M2 cells. The net result would be enhanced tissue repair and decreased inflammation. Created with BioRender.com.

## Data Availability

The data presented in this study are openly available in ZENODO at https://doi.org/10.5281/zenodo.7388941.

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
