# Peer review of "Differential Modulation of Human M1 and M2 Macrophage Activity by ICOS-Mediated ICOSL Triggering"

_ijms, 2023, doi:10.3390/ijms24032953_

Round 1

Reviewer 1 Report

Manuscript ID: IJMS-2119980

Title: Differential modulation of human M1 and M2 macrophage activity by ICOS-mediated ICOSL trigerring.

Remarks to the authors: 

The present study of Gigliotti et al. aims to characterize the effects of ICOS-ICOSL interaction on human macrophages. For this purpose, they used a recombinant soluble ICOS (ICOS-Fab2) to stimulate human monocyte-derived macrophages. Authors evaluated cytokines production and the cell migration induced in response to ICOS-Fab2. The paper is easy to read and rationally constructed. Results are well described but no explanation or conclusion has been made. Some points need to be improved to support the results. 

Comments:

Typo issue in abstract (line 22) and in results CLL7 (line 211)

Too much details about macrophages in cancer/tumor are presented in introduction but no cancer in results (microbial stimulation) or in discussion (role in anti-infectious defenses)

Some figures are not easy to read. Indeed, the representation in % is not the standard used.

Authors show too much group of stimulation with no real effect. So the take home message is diluted…

No conclusions have been made about modulation of cytokine production, surface markers, migration…

Figure 1: the various panel (right and left) of the flow cytometry data (Fig1e-f) are not explained (isotype control, unstimulated, stimulated with ?? ) 

The dot plot representation is not adapted to appreciate a slight difference of ICOSL expression. A more convincing graph should be used like overlapping histograms.

Authors try to discuss the modulation of ICOS but a few sentences before, they said that ICOS is not induced… it’s confusing.

Table1/Figure 2:

Authors should present the results of CD86/80 as mentioned in the text.

Histogram that summarized the table 1 would be appreciated. 

The % is not relevant. Authors must show the concentration of the cytokine in the graph.

Figure 3 and Figure 4: same comment about the concentrations

In legend authors talk about effect of ICOS-Fc ???

Figure 5: Authors should show the number of cells or the % of cells that have already migrate in each condition. Not only the % of control condition.

M&M: Line 351 “by immunofluorescence and flow cytometry” cytometry data have been presented but what do the authors mean by immunofluorescence ?.

Reviewer 2 Report

In this manuscript, the authors show that the activity of human M1 and M2 macrophages modulated by ICOS-mediated triggering of ICOSL differentially regulates cytokine secretion and cell migration. Their results suggest that this modulation of M1 and M2 cell activity promotes tissue repair, and hence induces anti-inflammatory effects. This manuscript includes some interesting findings regarding ICOSL-mediated M1 and M2 macrophage function, but the significance of each experiment is difficult to understand. In addition, as noted in comment 11, if the expression of ICOSL is different between human and mouse macrophages, the expression of ICOSL in macrophages from human samples should be confirmed. Therefore, this reviewer feels that this manuscript is not suitable for IJMS.

Major points 

1.     In line 62, the authors state “ICOSL relies on its ability to bind to osteopontin (OPN)”, but they do not explain what the important role played by the binding of ICOSL to OPN is. Furthermore, the authors use OPN in Figure 5, but as there is no explanation of OPN, the significance of its use is not clear. The authors should explain these points in the manuscript.

2.     In the authors’ experiments, LPS, LPS+IFNγ, or IFNγ was used to activate M1 cells, and LPS, LPS+IL-4, or IL-4 was used to activate M2 cells. What are the differences of these three types of stimuli for M1 and M2 cells? Do they result in different degrees of activation or intracellular signaling? In particular, M2 cells show different expression levels of IOSL depending on the stimulus, but is this difference associated with the degree of activation?

3.     In Figure 1, the authors analyze the expression of ICOS and ICOSL. Does the presence of ICOS-Fab2 change the expression of ICOSL, as you shown in Figures 2 to 6?

4.     In line 137, the authors mention the levels of CD80, CD163, and CD86, but it would be easier for the reader to understand if they also explained why these were analyzed.

5.     In Table 1, the authors analyze the secretion of several cytokines/chemokines, but it would be easier for the readers to understand the significance of the experiment if they explain why they analyzed these cytokines/chemokines.

6.     In line 175, the authors state “When IFNγ was added to these mixtures”. What does the addition of IFNγ induce in M1 cells? Regarding comment 2, what is the difference between LPS treatment and LPS+IFNγ treatment in terms of M1 cell activation?

7.     Similarly, the authors mention “LPS+IL-4 treated M2 cells” in line 189. What difference is induced in M2 cells between LPS and LPS+IL-4 treatment? What differences are responsible for the changes in cytokine/chemokine secretion in M2 cells?

8.     In the cell migration assay, the authors used CCL7 and OPN in their analysis, but it would be easier for the reader to understand if they state why these were used.

9.     In Figure 6, the authors show that ICOS-Fab2 treatment alters β-Pix expression in M1 and M2 cells, but does it also alter FAK phosphorylation and paxillin phosphorylation? In addition, does the expression of integrin change? These points should be explained.

10.  In line 283, the authors state that “ICOS-Fab2 decreased secretion by about 60% in M1 cells, while it increased it by about 2 folds in M2 cells”, when M1 cells were treated with LPS and M2 cells were treated with LPS+IL4. Please add an explanation that IL-6 is not increased in M2 cells by LPS treatment.

11.  In line 320, the authors state “also ICOSL expression displays an opposite pattern in human vs mouse macrophages”, but do the authors actually observe higher ICOSL expression in M2 cells compared with M1 cells in human clinical samples or in a database? This point needs to be clarified.

12.  In line 322, the authors state “ICOSL function may be influenced by its expression levels”. The authors should explain how the downregulation of ICOSL expression upon LPS treatment or LPS+IL-4 treatment of M2 cells relates to ICOSL function.

Minor point

Figures 1e, 1f, and Figure 4 do not appear in the text, and should be included.

Reviewer 3 Report

 Activated T cells express ICOS that, upon binding to its ubiquitously expressed ligand (ICOSL), regulates the immune response and tissue repair. The authors sought to determine the effect of ICOS:ICOSL interaction on human M1 and M2 macrophages in this study. They demonstrated that ICOS-Fab2 treatment regulated the secretion of cytokine, chemokine, and cell migration in M1 and M2 cells. Moreover, ERK1/2 and β-Pix participated in ICOS-Fab2-mediated signaling pathways.

There are many aspects should be clarified.

1.      It is unclear whether the ratio of M1 or M2 cells in the MDMs after treatment with GM-CSF or M-CSF. Please provide the data of CD80/CD86-positive or CD163/CD86-positive cells after treatment with GM-CSF or M-CSF in MDMs.

2.      Similar ICOSL+ M2 cell ratio after LPS or LPS+IL4 treatment were revealed in figure 1. However, CCL3 secretion was much lower in LPS+ICOS-Fab2-treated M2 cells than LPS+IL-4+ICOS-Fab2-treated M2 cells in figure 4. Regarding to ICOSL triggering cytokine secretion in macrophages, why CCL3 secretion was not associated with ICOSL expression in M2 cells?

3.      In figure 6, the phosphorylation ratio of ERK1/2 should be compared with total ERK1/2 not b-actin.

4.      The mechanisms of cytokine secretion and cell migration in ICOS-Fab2-treated cells are too superficial. Although ERK1/2 was reported in ICOS-Fc-mediated ECs and tumor cell migration, ERK1/2 participates in regulating multiple functions in different cells. The same phenomenon should be confirmed in macrophages in this study.

5.      The authors claimed that ICOS-Fab2 treatment increased the migration rate of macrophages, which was associated with enhanced ERK phosphorylation. However, M2 cell migration was inhibited by ICOS-Fab2 treatment, which was not associated with ERK phosphorylation status. Please discuss it in the manuscript.

6.      It is interesting to know which signaling pathways were participated in ICOSL-mediated cytokine secretion in M1 or M2 macrophages?

7.      The clinical relevance in this study was uncertain.

Round 2

Reviewer 2 Report

The authors have responded to most of my comments. However, there are still some concerns regarding the manuscript text, as follows.

1.     In Major point 10, “ICOS-CH3 decreased secretion by about 60% in M1 cells, while it increased it by about 2 folds in M2 cells”, I understand that this refers to the effect of ICOS-CH3 and does not refer to the absence of ICOS-CH3 in Figure S2. What is the 2-fold increase in secretion from M2 cells being compared to? Please describe this clearly so that the readers can understand it. In addition, I cannot understand the last part of the authors’ response to this comment, specifically, the part following “possibly because”. Could the authors please try to explain it again more clearly?

2.     In Minor point, “Figures 1e, 1f, and Figure 4 do not appear in the text, and should be included”, I did not mean that the panel is not visible, but I meant that these figures are not referred to in the text. In the revised version, the authors refer to  “e-f” in the text (line 125), but it would be better to write  “Figure 1e,f”. The text still does not refer to Figure 4, so the authors should do so.

3.     In line 123 of the revised version, “c-d” should be “b-d”.

Author Response

The authors have responded to most of my comments. However, there are still some concerns regarding the manuscript text, as follows.

  1. In Major point 10, “ICOS-CH3 decreased secretion by about 60% in M1 cells, while it increased it by about 2 folds in M2 cells”, I understand that this refers to the effect of ICOS-CH3 and does not refer to the absence of ICOS-CH3 in Figure S2. What is the 2-fold increase in secretion from M2 cells being compared to? Please describe this clearly so that the readers can understand it. 

According to the suggestions of the reviewer, the discussion of these data has been extended in the revised version of the Discussion, as follows: “Specifically, IL-6 secretion was 8-fold higher in M1 cells activated with LPS than in similarly activated M2 cells; and it was 36-fold higher in M1 cells activated with LPS+IFNγ than in M2 cell activated with LPS+IL-4. Furthermore, ICOS-CH3 decreased IL-6 secretion by about 60% in M1 cells activated with LPS, while it increased it by about 2.5 folds in M2 cells activated with LPS+IL-4”.

In addition, I cannot understand the last part of the authors’ response to this comment, specifically, the part following “possibly because”. Could the authors please try to explain it again more clearly?

IL-6 production was 1800 pg/ml in M2 cells activated with LPS alone and 560 pg/ml in those activated with LPS+IL4, which is in line with the higher production of TNF-a by the former cells (129 vs 51 pg/ml). Treatment with ICOS-CH3 did not change IL-6 secretion in M2 cells activated with LPS alone but it increased it by 2.5 folds in those activated with LPS+IL-4, reaching levels (about 1300 pg/ml) close to those of the former cells. One possibility is that IL-6 production in LPS-activated M2 cells is maximal and cannot be increased by ICOS-CH3, whereas ICOS-CH3 can increase IL-6 secretion in M2 cells activated with LPS+IL-4. This is just a speculation and we believe that it does not deserve to be discussed in the manuscript.

  1. In Minor point, “Figures 1e, 1f, and Figure 4 do not appear in the text, and should be included”, I did not mean that the panel is not visible, but I meant that these figures are not referred to in the text. In the revised version, the authors refer to  “e-f” in the text (line 125), but it would be better to write  “Figure 1e,f”. The text still does not refer to Figure 4, so the authors should do so.

The references to the Figure 1 panels have been ameliorated in the text and the reference to Figure 4 has been added.

  1. In line 123 of the revised version, “c-d” should be “b-d”.

The references to the Figure 1 panels have been ameliorated in the text.

Reviewer 3 Report

The authors did not detect an increased phosphorylation of total ERK. They changed the terms “ERK phosphorylation” with “expression of phospho-ERK”. However, Phosphorylation is the mechanism of regulating protein function and transmitting signals throughout the cell. Signal transduction results in a desired cellular response. Therefore, ERK1/2 phosphorylation status (phospho-ERK1/2 compared with total ERK1/2) is used to evaluate the activation of ERK1/2 signaling. The detection of phospho-ERK without total ERK is not acceptable. The author previously reported heterogeneous effect of ICOSL triggering in different cell types, since it inhibits expression of phospho-ERK in ECs but not in tumor cells or DCs. If total ERK1/2 antibody is not available. I would suggest to evaluate the activity of dephosphatase in this study.

Author Response

The authors did not detect an increased phosphorylation of total ERK. They changed the terms “ERK phosphorylation” with “expression of phospho-ERK”. However, Phosphorylation is the mechanism of regulating protein function and transmitting signals throughout the cell. Signal transduction results in a desired cellular response. Therefore, ERK1/2 phosphorylation status (phospho-ERK1/2 compared with total ERK1/2) is used to evaluate the activation of ERK1/2 signaling. The detection of phospho-ERK without total ERK is not acceptable. The author previously reported heterogeneous effect of ICOSL triggering in different cell types, since it inhibits expression of phospho-ERK in ECs but not in tumor cells or DCs. If total ERK1/2 antibody is not available. I would suggest to evaluate the activity of dephosphatase in this study.

Unfortunately, no spared lysates are available from those experiments and the WB sheets are too old to be reblotted. However, since data on ERK were ambiguous (as stated in the old manuscript) and the only other cell model where we previously showed effects on ERK was tumor cells, we decided to eliminate these data and focus on beta-Pix showing data that are consistent in M1 and M2 cells and fit with our previous data on dendritic cells. Therefore, Figure 6, Results and Discussion have been modified consequently.

Round 3

Reviewer 3 Report

The work has been improved and it is suitable for publication.